# Multiple sources of volunteered geographic information strengthen holistic estimates of lake visitation

Rachel M. Fricke[1]*, Spencer A. Wood[2], Julian D. Olden[1]

1 School of Aquatic and Fishery Sciences, University of Washington, Seattle, Washington, United States of America, 2 eScience Institute, University of Washington, Seattle, Washington, United States of America

* rachel.m.fricke@gmail.com

## Abstract

Lakes provide human societies with a wide range of cultural ecosystem services (CES), yet these benefits are rarely quantified. Site visitation is frequently used to assign CES values to recreational destinations, but traditional approaches for estimating lake visitation have limited spatiotemporal extent. Visitation estimates increasingly leverage volunteered geographic information (VGI) to address this challenge. We compared the utility of five different sources of VGI from Flickr, eBird, iNaturalist, Twitter, and Gaia GPS, which broadly encompass lake users with different motivations for interacting with nature. We evaluated the potential for predicting on-site visitation from in-person counts by testing models informed by unique combinations of VGI sources at urban and suburban lakes in western Washington. Additionally, we investigated the amenities driving differences in relative lake visitation by modeling visitation as a function of lake attributes (e.g., tree cover, water quality, built infrastructure). All VGI sources were included in the top-performing visitation models, suggesting they provide significant and unique contributions to estimates of overall lake use (*combined $R^2$ = 0.85*, in-sample testing). Given that these VGI sources reflect different types of lake users seeking unique CES, we conclude that holistic VGI visitation estimates should incorporate a diversity of VGI sources. Our results also reveal that built lakeside infrastructure is the predominant driver of visitation at lakes in western Washington, suggesting that spatially equitable updates to amenities will encourage public lake use. We urge greater consideration of the accessibility of different lake-based CES across the landscape and among diverse communities in future lake recreation planning, and suggest that VGI-based estimates of lake visitation offer a robust way to inform this process.

**Data availability statement:** All shareable data underlying the results presented in this study are available on Zenodo [10.5281/zenodo.17654557]. VGI data cannot be shared publicly because the data are owned by third parties and authors do not have permission to share the data. VGI data related to this study, including volunteered geographic information records from Flickr, Twitter (now X), Gaia GPS, eBird, and iNaturalist, are available upon download or request from the following webpages and application programming interfaces (APIs): Flickr [https://www.flickr.com/services/developer/api/], eBird [https://science.ebird.org/en/use-ebird-data/download-ebird-data-products], iNaturalist [doi: 10.15468/ab3s5x], and Twitter (X) [https://developer.x.com/en/docs/x-api]. Please note that the free Academic API the authors used to acquire Twitter (X) data is now deprecated under new company policies. Researchers interested in accessing Gaia GPS data must submit a formal data access request directly to the company outlining their research goals and ensuring compliance with relevant privacy regulations [https://www.gaiagps.com/].

**Funding:** This work was supported by a U.S. Geological Survey Northwest Climate Adaptation Science Center award G17AC00218 to Rachel Fricke, the Future Rivers program at the University of Washington as part of a NSF National Research Traineeship award (DGE 1922004), and the University of Washington eScience Institute. The funders had no role in study design, data collection and analysis, decision to publish, or preparation of the manuscript.

**Competing interests:** The authors have declared that no competing interests exist.

## Introduction

Ecosystem services represent a valuable lens by which to understand the diverse benefits humans derive from the natural world. Freshwater ecosystems, such as lakes and reservoirs, offer drinking water, support biodiversity, and provide numerous types of recreation opportunities (e.g., fishing, boating, swimming, camping), all of which are highly valued by people [1,2]. Additionally, non-material cultural ecosystem services (CES) such as sense of place, physical and mental health, and spiritual or aesthetic value are supported, despite being markedly undervalued both globally and in freshwater ecosystems, including lakes [3,4]. The value of water-based services varies spatially within and between watersheds, highlighting that understanding the spatiotemporal distribution of freshwater CES is critical to managing water resources with the goal of meeting and preserving societal demands and values [5,6].

Lakes and reservoirs are globally ubiquitous and support critical ecosystem functions, provide numerous goods and services, and contribute to sustainable local and regional communities [1,7]. Furthermore, lakes (and their adjacent parks) in urban and suburban settings serve as hotspots of connections to nature and offer heat stress relief in the midst of urban heat island effects and more intense and frequent climate-induced heatwaves [8]. Given lakes' multidimensional role in supporting human well-being, understanding human activity on them is important for natural resource valuation, prioritizing restoration and conservation efforts and ensuring equitable access for people [9,10]. This necessity only heightens when considering the magnitude of human impacts on freshwater ecosystems, both today and projected in the future [11–13].

Human visitation is one fundamental metric by which researchers commonly assess CES at sociocultural points of interest [14]. Human activity on waterbodies across time and space is frequently inferred from sparsely conducted visitor counts providing data with limited spatiotemporal scope [15]. Conventional approaches typically rely on direct in-person surveys or passive sensors for counting people or vehicles (often at boat launches), while other studies rely on mail-in questionnaires sent to registered anglers or boaters [16]. All of these methods offer only a snapshot estimate of activity at particular locations and times [17,18]. Furthermore, such approaches are often biased toward older individuals and those who frequently visit sites of interest, and participants can typically only recall the locations they have visited most recently [19,20].

Visitation data is valuable for understanding human preferences in leisure and recreational choice. High quality visitation estimates allow managers to identify hotspots of recreational activity and assess the drivers (e.g., environmental quality, built infrastructure) of revealed preferences in where humans engage in leisure and activities [21]. In a lake context, these can be characterized as lake users who benefit from spending time on or adjacent to lakes and the supply of different types of lakes or lake access available within a given region they might choose to visit. From a resource planning perspective, revealed preference analyses enable managers to assess which attributes of sites attract visitors [22,23] and prioritize enhancement of these features when upgrading existing or developing new points of access [24].

More recent work has modeled lake visitation and overland connectivity from volunteered geographic information (VGI) derived from mobile device applications developed specifically for anglers [25–27]. However, such methods narrowly target specific user groups and associated activities (e.g., anglers and boaters), precluding opportunities for evaluating CES in locations with a diversity of activities. With increasing internet connectivity and mobile device use, researchers have begun testing whether mobile device applications and crowdsourced data generated by user posts can serve as proxies for empirical visitation at recreational sites of interest [28]. For example, Keeler et al. [29] and Nelson et al. [30] observed the number of geotagged photographs on the image-sharing website Flickr had utility for estimating on-site lake visitation across regional scales. Although data from diverse mobile device applications are commonly all labeled VGI, and some are at times considered as substitutable, different mobile device applications have distinct user bases and community cultures [31].

VGI is often strongly associated with ground-truthed visitation estimates for recreational points of interest. At US National Parks, for example, estimated visitation by the National Park Service each month corresponded highly with the number of photographs shared on Flickr of the same months [32]. Significant positive associations between estimates of actual visitation and those estimated from VGI have also been quantified at local city parks, regional state parks, and global recreational sites [33]. Similar analyses have been conducted for water bodies, but thus far lake visitation has exclusively been evaluated with data from singular VGI sources, predominantly Flickr [29,30,34]. Ideally, estimates of visitation incorporate multiple sources of mobile data, as data from different types of VGI sources (e.g., social media, citizen science, activity sharing) represent specific user groups and their coverages vary in temporal and spatial granularity [35,36]. The value and accuracy of different VGI sources for estimating lake visitation has yet to be evaluated, despite emerging evidence for the benefit of such investigations [37,38].

Here, we use data from five different sources of VGI – representing users with potentially different motivations to interface with nature – to model visitation at lakes in Western Washington, United States. These VGI are derived from mobile device applications that passively collect geo-referenced records from users who interact with the applications for different purposes that may broadly reflect different types of human-nature interactions [39]. Our study assesses the relative value of different sources of VGI for estimating visitation by comparing predictive visitation models informed by various combinations of different VGI data sources to empirical visitation counts. We also seek to better understand which built and environmental attributes (e.g., tree cover, water quality, built infrastructure facilitating lake access) may be responsible for differences in lake visitation to inform how urban planners might meet leisure and recreation preferences and enhance the delivery and durability of CES.

## Methods

### Study Area

The study area encompasses 50 urban and suburban lakes in Western Washington, United States (Fig 1). The lakes range in surface area from 41 to 180 km². Water clarity of the lakes span from mesotrophic (moderate nutrients, moderate water clarity) to oligotrophic (low nutrients, high water clarity) (mean Secchi depth = 3.7 m, SD = 1.6 m). Every lake has at least one public access point (e.g., park, beach, boat launch), and provides recreational fishing opportunities typically involving stocked or established populations of rainbow trout, yellow perch, black basses, and sunfishes.

### Modeling Visitation at Lakes

**Volunteered Geographic Information.** We evaluate the utility of geographic information from five VGI sources representing two nature-based citizen-science observations (eBird, iNaturalist), an image-sharing application (Flickr), a microblogging site (Twitter, now X), and a mapping application (Gaia GPS). eBird is a citizen science project with a mobile platform that allows users to share a variety of information about birds including observations and photographs. iNaturalist is a similar platform which encourages users to document all species of flora and fauna. Both eBird and

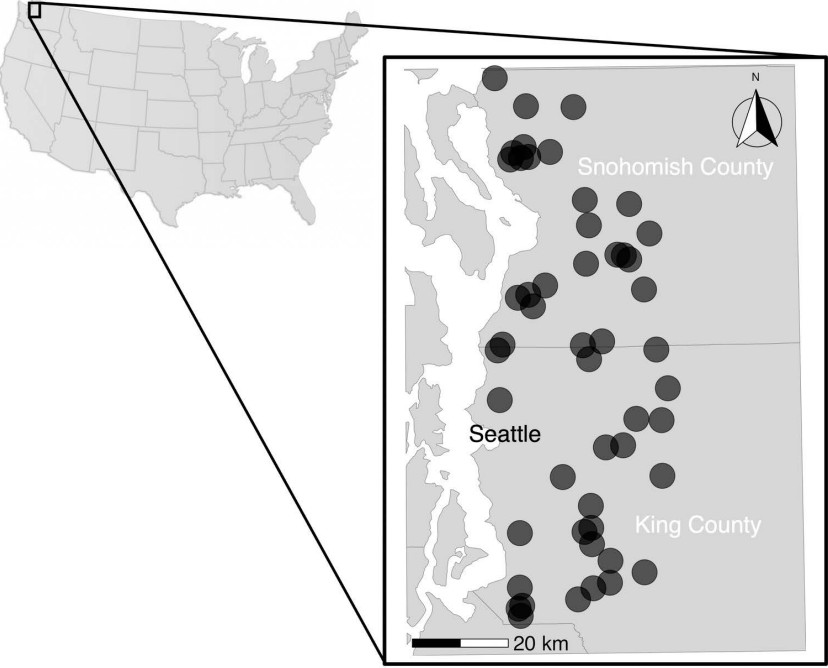

**Fig 1. Locations of 50 lakes in western Washington, United States.** The basemap is freely available from the US Census Bureau.

iNaturalist users generally have active interest in the existence and conservation of living species [40]. Flickr is an image-sharing application for amateur and professional photographers reflecting users' aesthetic appreciation of natural spaces, Twitter is a widely used micro-blogging site through which users can also share images and videos, and Gaia GPS is an activity tracking platform on which users track their runs, bike rides, and other forms of physical activity [28,37,41]. When selecting different VGI platforms we navigated tradeoffs between large, geographically dispersed data, publicly and easily available datasets, and potential platforms being used based on common lake activities. If we want to use VGI platforms as valid indicators of visitation they need to be easily and readily available – managers do not have the luxury of spending years acquiring data that is not publicly available. These five platforms were selected largely because the data are accessible to researchers (e.g., through public acquisition interfaces or data sharing agreements, described below). Four of the platforms have also been the primary focus of previous studies of competing methods for estimating visitation (see reviews by Ghermandi et al., Wilkins et al.) [33,42].

The analyses involve a five-year period from 2015−2019 (inclusive) because this is the period of time for which on-site visitation data was available for the greatest number of lakes. This timeframe also falls well after the launch of all five VGI sources, and before the onset of the COVID-19 pandemic. Records from all of these sources include anonymous user identifiers and the geolocation (latitude and longitude) as metadata. We obtained posts for all VGI sources for the years 2015–2019 through various means. Flickr posts were obtained by spatially querying the Flickr application programming interface (API) with lake polygons. Twitter posts were acquired by querying the Twitter API v2 with lake centroid coordinates and the longest radius required to encapsulate the whole lake, then further spatially filtering acquired posts to lake polygons. All spatial processing of geotagged VGI included the lake and a 50-m buffer extending beyond the lake perimeter. This buffer distance was chosen because on-site counts only include visitors on the lake and shoreline; thus, a relatively small buffer excludes lake-adjacent park users who may not have been engaging with lake-specific benefits. This buffer distance is comparable to previous studies that spatially filtered geolocated social media records to lakes and their shorelines [26,29].

eBird and iNaturalist records were downloaded from their respective online data access portals for King and Snohom-ish counties [43]. Gaia tracks were provided as anonymous lake-level daily summaries of activity through a data sharing agreement with Outside, Inc. These daily counts were inferred from the overlap of breadcrumbs (the geospatial tracks created by Gaia users) with lake polygons. All VGI data collection and analysis methods complied with the terms and conditions for each respective data source. Next, we spatially filtered user posts from eBird, iNaturalist, Flickr, and Twitter to lake polygons from the USGS National Hydrography Dataset [44]. We then aggregated posts by VGI source, user ID, lake, and day to calculate the number of user-days – or distinct users who visit per day – per VGI source at each lake [28]. In other words, if a user posted to the same platform from the same lake multiple times in one day this is considered a single lake user-day. This aggregation was done to prevent users who posted to a VGI source multiple times succes-sively from a single location (i.e., high engagement) from dominating the analysis in comparison to users who post less frequently, but nevertheless are visiting lakes.

**On-site Lake Visitation.** On-site estimates of lake visitation come from counts carried out by trained volunteers for King County Department of Natural Resources and Parks and Snohomish County Conservation and Natural Resources. Volunteers (typically lakeside residents) counted the number of boaters, swimmers, and other recreationists on the lake water and shoreline at instantaneous points in time, bi-weekly from May through October, over the years 2015–2019. Counts were collected at various times of day between 8:00 and 18:00 and on random days of the week. Observations were made in addition to volunteers' primary purpose to measure water quality metrics [45,46], so lakes were selected based on the counties' priority sites for water quality monitoring, as well as volunteer availability at each lake. Volunteers were instructed to collect counts at roughly the same time of day and day of week at the assigned lake, but upon analysis of the volunteer dataset we found that the dates and times of counts were inconsistent both within and between different lakes. To relate our on-site dataset to VGI counts, we temporally aggregated the on-site visitation estimates as annual mean measures of the instantaneous counts by lake. Given the sparsity and inconsistency of total counts between years and lakes, we averaged instantaneous counts on a yearly basis to dampen potential biases of individual outlying counts while still providing a meaningful representation of lake visitation.

**Visitation Model.** We compared the utility of different suites of VGI datasets for reflecting on-site lake visitation using linear mixed-effects models. The analyses modeled the annual average on-site visitation (from instantaneous volunteer counts) as a function of the fixed effects describing annual cumulative user-days for different combinations of VGI sources and a random lake effect to reflect variability among lakes. Previous work comparing VGI regression approaches found little difference between standard major axis (SMA) regression and ordinary least squares (OLS) algorithms [47]. Furthermore, SMA regression cannot be done with more than one predictor variable, and thus we proceeded with OLS to predict on-site visitation rates. Model performance was assessed with Akaike's information criterion (AICc) and calculating the delta AICc for each candidate model. Next, we evaluated the top visitation models' ability to predict visitation at lakes lacking on-site data with out-of-sample testing. For each top model, we trained the model on all observations from two-thirds of the lakes in our dataset and tested it over all observations for the remaining one-third of lakes, repeated for 1,000 estimates. Given discrepancies in the number of total years each lake was sampled over the five years of our study (e.g., some lakes were only sampled for two or three years while others were sampled for all five), we grouped cross-validation by lake to ensure accurate representation. In other words, all years of data from a single lake were included in either the training or test dataset. This approach also allowed us to assess how effective a model trained on one set of lakes was at predicting visitation on an entirely different set of lakes. We calculated the average root mean squared error (RMSE) and R-squared of the out-of-sample tests to assess the models.

## Environmental and Built Infrastructure Influences on Visitation

**Measures of Lake Attractiveness.** To assess the attractiveness of lakes to visitors, we collated information on public amenities, lake water quality, and tree cover in the public park and along the shoreline. We tabulated the presence or

absence of amenities on lake shorelines and in adjacent lake-parks (including parks, bathrooms, shelters, playgrounds, swimming beaches, docks, and boat ramps) from lake-park descriptions on King and Snohomish counties' websites [45,46]. We define parks as city, county, or state-managed parks and open green spaces, while swimming beaches are sandy shorelines with wading areas (these may or may not be roped off and have lifeguard supervision). Our water quality estimates are annual average Secchi depth (m), which was measured at the deepest point of lakes by county volunteers at the same time visitor counts were taken for the years 2015–2019. Secchi depth measures the transparency of water, where higher depth values correspond with clearer (and perceived "cleaner") water. Lastly, tree cover for the 50 m shoreline buffer and lake-park areas was calculated by overlaying polygons of lake shorelines and lake-adjacent parks with the National Land Cover Dataset Tree Cover raster from 2019 in ArcGIS and calculating the percent lakeside area with 50% or greater canopy cover [48]. These predictors were selected because they can reasonably be influenced by natural resource managers, and previous studies have demonstrated visitors are willing to travel further to lakes and parks which offer superior water quality and built and natural amenities [29,30].

**Revealed Preference Model.** We modeled visitation estimates as a function of lake amenities, water quality, and shoreline tree cover to assess the associations between measures of lake attractiveness and degree of human use. Visitation estimates came from the previously described visitation model which predicted visitation from all VGI sources at all 50 lakes for the five years of our study, thus also including years during which we lacked on-site data at certain lakes. Correlated lake attributes related to the presence of amenities (parks, bathrooms, shelters, public docks, playgrounds, and swimming beaches) were aggregated into a single lakeside amenity variable based on each of these predictors having a variance inflation factor (VIF) greater than 2.5 [49]. These six significantly correlated general amenities were combined into a lake amenity score metric with a range from one to six, while boat ramps were included as a separate presence/absence variable to reflect recreation specifically for boating and fishing. Ultimately, our linear mixed-effects model included the combined built lakeside infrastructure variable, boat ramps, water quality, shoreline tree cover, and a random lake effect as predictor variables, with visitation estimates as the response. All statistical analyses were completed using the lme4 package in R version 4.3.1 [50].

## Results

Across all 50 lakes, the number of total VGI user-days were as follows for the five years of our study: eBird ($n=1779$); iNaturalist ($n=224$); Flickr ($n=389$); Twitter ($n=1274$); and Gaia GPS ($n=604$). The mean number of total user days per lake across all VGI sources and years was 83.6, with a range from 1 to 1096. Green Lake – the most urban lake of our dataset in the city of Seattle, with a park spanning its entire shoreline – had the highest number of total user-days for all VGI sources except eBird (Fig 2). Many lakes had years in which there were zero records for one or more VGI datasets. User-days across most of the different VGI sources exhibited marginal to low cross-correlation with one another, but to varying degrees, with eBird and iNaturalist exhibiting the weakest correlation with other sources. The pairings of Flickr~Twitter ($r=0.53$), iNaturalist~Gaia ($r=0.50$), Gaia~Flickr ($r=0.42$), Twitter~Gaia ($r=0.31$), iNaturalist~Flickr ($r=0.28$), iNaturalist~Twitter ($r=0.26$), and iNaturalist~eBird ($r=0.26$) were significantly correlated ($p<0.001$) while Twitter~eBird ($r=0.18$), and Gaia~eBird ($r=0.15$) were also correlated ($p<0.01$ and $p<0.1$, respectively) (Fig 3). Flickr~eBird was the only VGI source pairing that was not correlated ($r=0.04$).

Associations between annual cumulative VGI user-days and on-site estimates of visitation were moderate. iNaturalist ($r=0.32$), Flickr ($r=0.28$), Twitter ($r=0.35$), and Gaia GPS ($r=0.32$) all positively correlated with on-site visitation, while eBird ($r=0.04$) showed little correlation (Fig 4).

In our comparison of alternative models using VGI sources to estimate on-site visitation, the model informed by Twitter and Gaia performed the best ($n=231$, *combined $R^2=0.845$*) with the VGI sources as fixed effects and lake identification as a random effect (Table 1). Both Twitter ($p=0.012$) and Gaia ($p=0.001$) significantly predicted on-site visitation in this model. The top-supported models are those with a delta AICc<2 and all VGI data sources were included, in different combinations, in the top-ranked performing models.

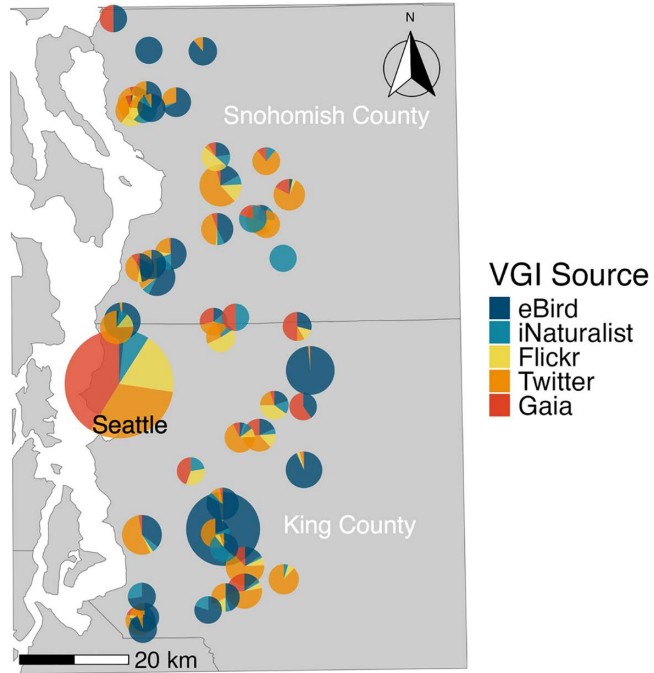

**Fig 2. Map of cumulative VGI user-days at 50 lakes in western Washington for 2015-2019 on eBird (dark blue), iNaturalist (teal), Flickr (yellow), Twitter (orange), and Gaia (red).** Point size corresponds to the number of user-days and point locations are jittered to ease interpretation. The base-map is freely available from the US Census Bureau.

We used the model informed by all five VGI datasets ($n = 231$, *combined* $R^2 = 0.85$, in-sample testing) to predict empirical visitation across all sites (and for years in which we lacked empirical data for certain lakes). Predicted mean annual visitation positively correlated with on-site visitation, though this model and the others we tested explained only a modest percentage of the data's variance in out-of-sample testing (Delta AICc = 4.62, RMSE = 1.06, $R^2 = 0.12$) (Fig 5).

The lake attributes we tested in our revealed preference model explained a significant proportion of variance in estimated visitation between lakes ($n = 250$, *conditional* $R^2 = 0.96$, *marginal* $R^2 = 0.28$) (Fig 6). The only significant coefficient in the model was the lake amenity metric, which tallied the presence/absence of parks, bathrooms, shelters, public docks, playgrounds, and swimming beaches. Water quality (Secchi depth), presence of boat ramps, and tree cover demonstrated no significant contribution to lake preference.

## Discussion

Our models incorporating data from multiple VGI sources are found to produce more accurate estimates of human visitation than models with any single VGI source alone. This is likely because different mobile device applications are used by different groups of people and therefore better capture the full range of lake users and activity types, and would therefore be a better reflection of the associated CES value. Our second analysis of preference for lake attributes indicates that built infrastructure supporting public amenities – such as playgrounds, parks, shelters, bathrooms, and swimming beaches – strongly promote lake use, whereas water clarity, boat ramps, and tree cover contribute less to visitation rates. These findings can aid resource managers in understanding lake visitation across urban-rural landscapes, help anticipate hotspots for lake degradation associated with human activities, and assist planners to ensure equitable access to lakes for different intended uses and associated ecosystem services.

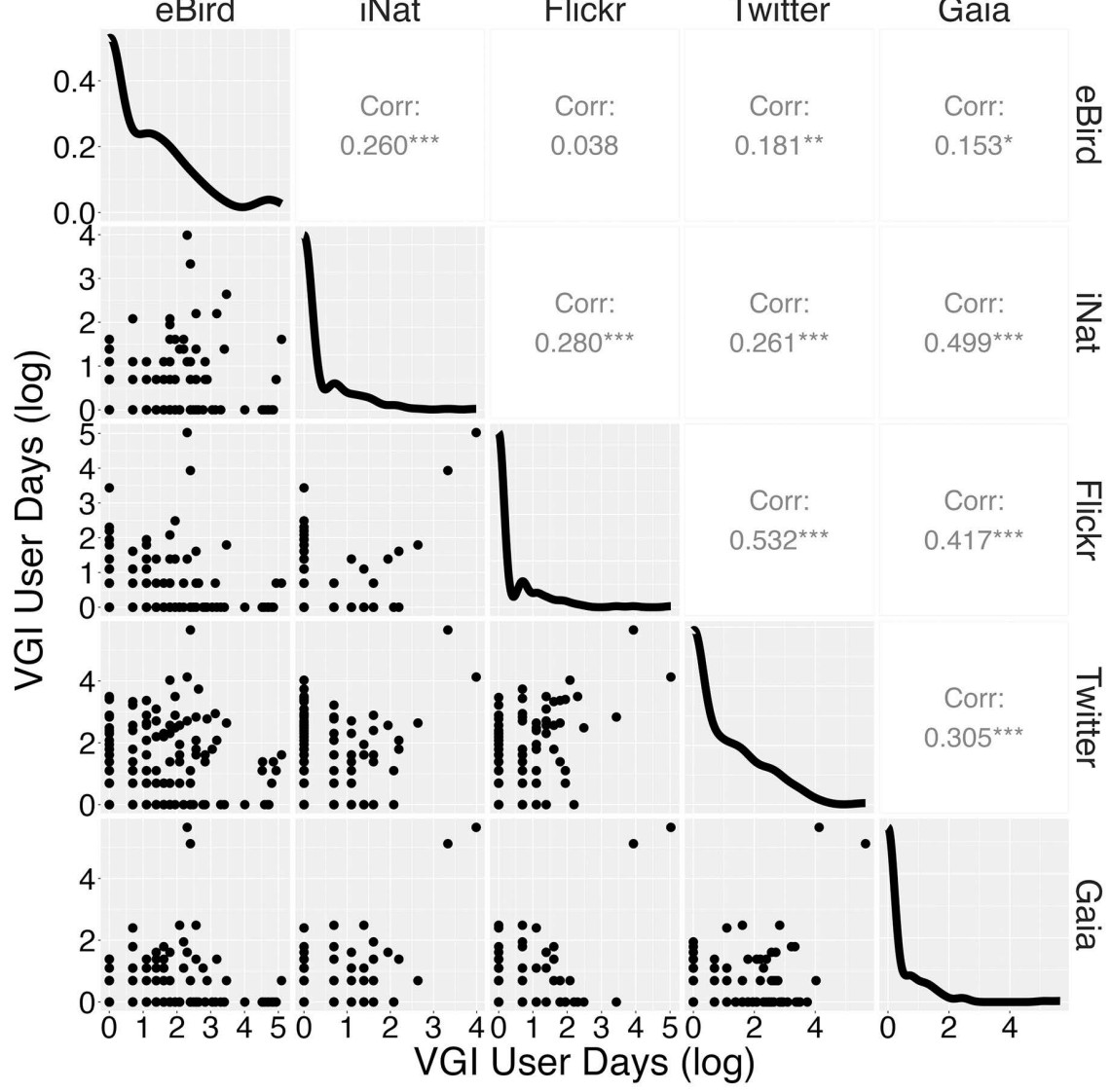

**Fig 3. Scatterplots (lower-left), density plots (diagonal), and Pearson's correlation coefficients (upper-right) comparing VGI datasets.** Statistical significance is indicated by *(<0.1), **(<0.01), and ***(<0.001).

### Benefits of Diversifying Volunteered Geographic Information Sources in Visitation Models

The visitation model informed by Twitter and Gaia was best at predicting on-site lake visitation in western Washington, based on model fit and cross-validation, followed closely by models that included at least one additional data source. All VGI sources in our analyses were included among the top-performing models. Previous studies of public lands across the United States and Canada have similarly found that no single VGI data source outperforms others when modeling on-site visitation [51,52]. Wood et al. [36] found that visitation models with multiple VGI data sources are better at estimating visitation in cross-validation, and suggested that weak correlation of posting frequencies among VGI sources may indicate that each platform represents distinct user-groups participating in distinct recreational activities. Our results support this suggestion by highlighting generally weak associations between user-days

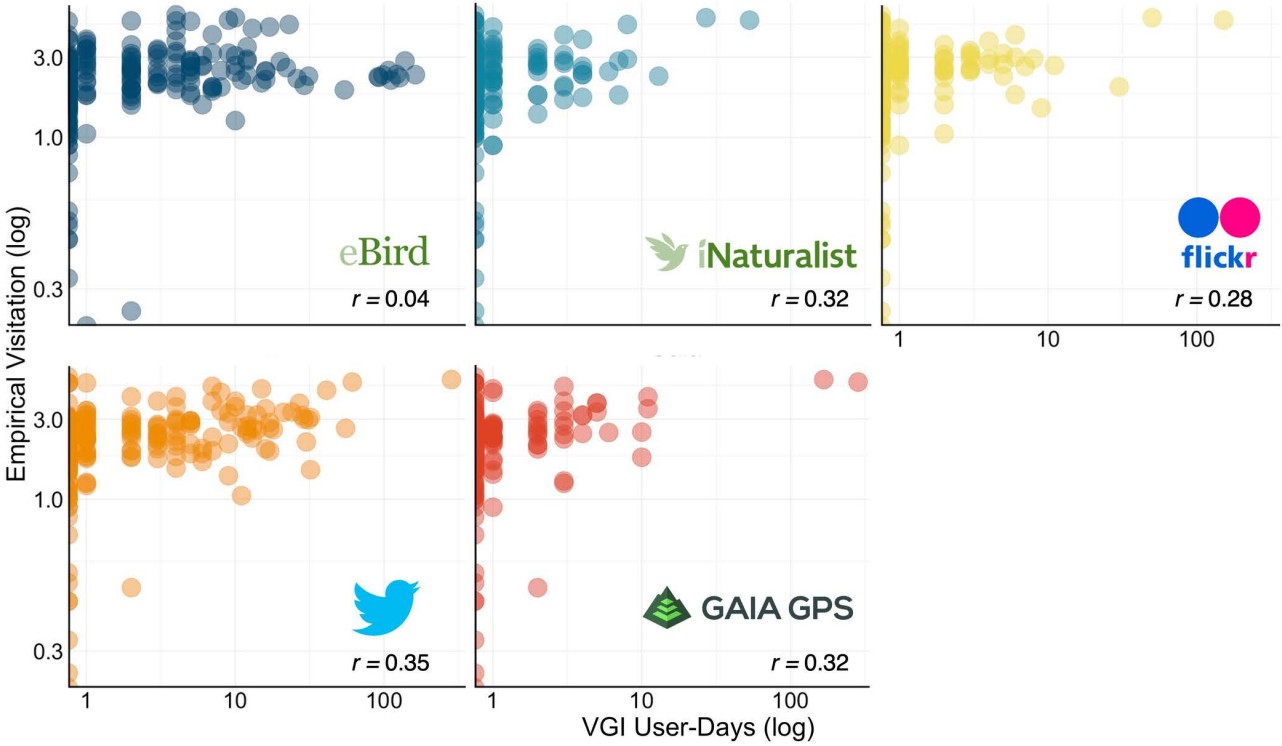

**Fig 4. Scatterplots of annual cumulative VGI user-days from eBird, iNaturalist, Flickr, Twitter, and Gaia GPS versus annual mean on-site visitation estimates.**

**Table 1. The top ten candidate visitation models relating mobile platform use and on-site visitation estimates. K is the number of parameters in the model, Delta AICc is the difference between AICc of the best fitting model and that of the top model, Model Likelihood is the relative likelihood, AICc Weight is the Akaike weight, and the R² and root mean square error (RMSE) values are from out-of-sample testing. All models with Delta AICc < 5 are listed.**

| Model | K | AICc | Delta AICc | Model Likelihood | AICc Weight | R² | RMSE |
|---|---|---|---|---|---|---|---|
| Twitter + Gaia | 6 | 393.4 | 0.00 | 1.00 | 0.17 | 0.14 | 1.09 |
| iNat + Twitter + Gaia | 7 | 394.8 | 1.35 | 0.51 | 0.08 | 0.13 | 1.08 |
| eBird + Twitter + Gaia | 7 | 394.8 | 1.37 | 0.50 | 0.08 | 0.13 | 1.07 |
| Flickr + Twitter + Gaia | 7 | 394.9 | 1.47 | 0.48 | 0.08 | 0.11 | 1.09 |
| iNat + Twitter | 6 | 394.9 | 1.50 | 0.47 | 0.08 | 0.12 | 1.02 |
| Gaia | 5 | 395.7 | 2.30 | 0.32 | 0.05 | 0.07 | 1.08 |
| eBird + Flickr + Twitter + Gaia | 8 | 396.3 | 2.93 | 0.23 | 0.04 | 0.10 | 1.08 |
| eBird + iNat + Twitter + Gaia | 8 | 396.3 | 2.93 | 0.23 | 0.04 | 0.12 | 1.08 |
| Flickr + Gaia | 6 | 396.3 | 2.94 | 0.23 | 0.04 | 0.04 | 1.07 |
| iNat + Flickr + Twitter + Gaia | 8 | 396.4 | 2.99 | 0.22 | 0.04 | 0.11 | 1.10 |

estimated from different VGI sources for lakes, particularly those expected to have very different user-bases (e.g., eBird and Twitter).

Our analysis suggests that including a diversity of VGI datasets rather than singular sources can benefit predictions of on-site lake visitation. What remains a persistent challenge is understanding how the use of different mobile device

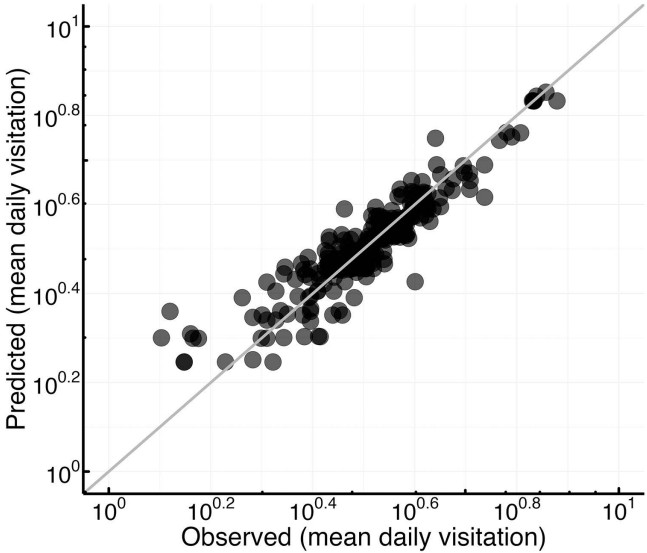

**Fig 5. Observed and predicted in-sample annual mean daily visitation at lakes, predicted as a function of annual VGI user-days from eBird, iNaturalist, Flickr, Twitter, and Gaia.** Predicted values are plotted relative to observed empirical visitation ($R^2=0.89$), and the slope line indicates a 1:1 relationship.

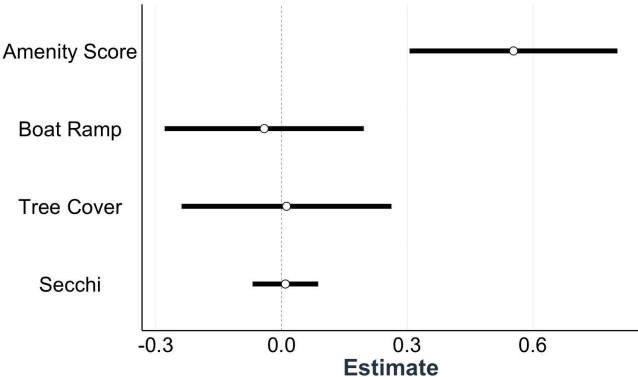

**Fig 6. Coefficient estimates from the revealed preference model relating lake visitation to lake attributes.** White circles represent means and bars are 95% confidence intervals of the estimates.

applications, and therefore the volumes of VGI data from different platforms, are related to specific CES for people. Havinga et al. [37] proposed a framework of CES service categories (e.g., activity, aesthetic, artistic, knowledge) and how each is associated with specific types of VGI. According to the framework, Gaia GPS is an indicator of activity services because its users are physically interacting with the environment, whereas Flickr is an indicator of aesthetic appreciation or artistic inspiration, and eBird an indicator of connection to nature. However, there is also growing recognition that CES benefits may vary considerably within the user-group of a single VGI source [53]. VGI sources probably represent more than just one CES, and some CES are easier to recognize and classify through VGI analysis than others. Understanding representation of specific types of CES, or lack thereof, in selected data sources and across different ecosystems is critical for considering how ecosystem valuation using VGI analyses can support natural resource management and planning.

## Amenities Drive Lake Visitation

Our analysis of visitation as a function of lake attributes indicates that built lakeside amenities are the strongest driver of lake visits. This confirms previous findings that lakeside structures are a more powerful predictor of visitation than attributes such as water quality [30]. Though studies in the midwestern U.S. have associated improved water quality with increased lake visitation [29], the suite of lakes studied here do not vary substantially with respect to water clarity. Some lakes in our analysis have a history of infrequent algae blooms, however such events are rare and typically occur over only a few weeks and thus were not captured in annual average clarity that was calculated specifically to align with our empirical and VGI visitation data. Previous lake recreation studies have also recognized boat ramps as a driver of lake use [29], but our revealed preference model did not identify boat ramps as a significant variable predicting lake visitation. This is likely because the vast majority of lakes in our study have boat ramps and fishing docks, so angler and boater access is less of a limiting factor across lakes.

Tree cover has variable effects on visitation in the revealed preference model, highlighting the fact that some people may be attracted to undeveloped natural lakes to connect to nature while others may be attracted to developed lakes given the host of amenities they offer. For example, lakes can help alleviate the negative impacts humans experience from urban heat and noise [54], and the addition of canopy cover to existing green and blue spaces enhances evapotranspiration-based cooling influences of urban waterbodies [8]. At the same time, many people are drawn to recreational areas that support opportunities for children to play, picnic benches for socializing, and well-maintained access points for swimming and other activities [55]. Previous research has reported a similarly complex relationship between tree cover and visitation of urban parks inferred from VGI, suggesting that differences in behavior between users may play a role [56].

Managers face a challenge in determining how to balance demands for competing CES through lakeside development or restoration. While water-based activities contribute to human well-being, they can simultaneously impose stressors on aquatic systems such as depleting aquatic and riparian habitat quality, altering species' behaviors, and changing the biogeochemical cycles of aquatic ecosystems [57,58]. Negative impacts to the natural environment subsequently adversely affect nature-based activities such as birdwatching and aesthetic appreciation. These competing demands are somewhat addressed by the heterogeneous spatial distribution of public activity-specific infrastructure (e.g., boat ramps, fishing docks, swimming beaches, natural preserves) at lakes, but access to lakeside environments supporting different lake-based CES is not equitable across the landscape. When allocating funds and resources to lakeside enhancement projects, managers should carefully consider tradeoffs between enriching built environments and introducing environmental stressors related to lake use hotspots (e.g., garbage, deteriorating riparian quality) which may impair CES derived from more natural lake environments [59]. Furthermore, intentional spatial zoning of public lake shorelines can help facilitate multiple types of CES that may directly conflict with one another, such as swimming and fishing [60].

## Limitations and Future Directions

Our study would have benefited from more accurate and abundant on-site visitor count data. Improved on-site data would not only increase confidence in the performance of our visitation models, but could also facilitate an analysis of absolute, rather than relative, visitation. Lake visitation is also highly seasonal, and a robust, year-round on-site dataset would have allowed us to assess the ability of VGI to estimate temporal patterns in human activities at lakes.

Utilizing instantaneous empirical data for our validation dataset presented a challenge. Given the sparse nature of our on-site and VGI data we needed to temporally aggregate both datasets to an annual basis, which reduced the already relatively small number of observations in our dataset. Typically, studies leveraging VGI to estimate visitation have relied on census counts of visitors collected with pedestrian or vehicle counters or other types of sensors at sites with controlled access points [61]. While this could be achieved at some lakes, by counting traffic at boat launches or other singular access points, it would be difficult to count every visitor to lakes which have numerous access points and can be reached via many modes of transportation. A study focused on the subset of lakes that do have controlled access would be biased

towards locations that are primarily accessed by vehicles and designed to serve primarily boaters and anglers. Previous studies have converted instantaneous counts into raw total visits or visitors per day to approximate the count estimates that would be produced by a passive sensor [62]. Unfortunately, we lacked the in-person survey or passively collected data to do so, and our study is more limited than previous work in this regard because we do not know how our estimates of relative visitation relate to the actual total number of people visiting a lake.

The VGI data sources used in this study are all convenience samples that are not representative of the population of lake users [28,35,37]. Visitors represented by some VGI sources, for example, are known to be younger [37] or biased toward a particular gender [35] compared with actual visitors. Furthermore, there are reasons why lake visitation may be challenging to measure using VGI, compared with studies investigating potential of VGI for terrestrial parks and protected areas. Individuals participating in water-based activities – such as fishing, boating, and swimming – may be less likely to engage with their mobile device while visiting a waterbody [28]. This may result in lower amounts of available VGI and, since VGI patterns tend to more accurately represent empirical visitation at sites with greater volumes of VGI [37], the need to aggregate data to annual or monthly scales, as in this study. It remains unclear what the utility of VGI will be for estimating lake visitation over shorter timescales [33]. Future research should explicitly show how specific sources of VGI are associated with CES through on-site surveys that categorize lake visitors by activity type and the value they receive from the visit.

Previous work has demonstrated discrepancies between the CES described in visitor use survey responses and CES inferred from automated analysis of text and images included in VGI, suggesting that further research is needed to understand how well VGI reflects visitors' perceived benefits at recreational sites of interest [63]. Here we argue that multiple sources of VGI better estimate visitation because more diverse user-groups are represented, but this is likely an oversimplification of the true CES the individuals that posted to mobile device applications are experiencing. Urban and suburban lakes offer numerous benefits to humans that may not be fully reflected in any source of VGI, such as improving health, reducing stress, providing social and place-based belonging, and mediating negative impacts of urban heat and noise [3,64]. A comprehensive in-person survey asking lake users which CES they relate to through their VGI posts and which benefits they feel are not captured by their mobile device application activity would further refine the capabilities and limitations of VGI for estimating lake-based CES. Moreover, given that VGI based approaches may poorly represent specific socio-demographic groups of people, holistic representation of lake users in CES assessments may be best achieved through a combination of measurement techniques including VGI, surveys, and in-person workshops [65].

## Practical Implications

Managers responsible for open-space planning would ideally have access to data from a wide range of VGI sources that holistically represent potential CES associated with recreation over time. However, there are practical and ever-evolving limitations to VGI acquisition. Notably, access to VGI is constantly evolving [38,66]. For example, geolocated Instagram posts are no longer available through an API, and since the completion of our analyses Twitter has changed ownership and the new company (X) has placed burdensome costs and rate limits on its API. Furthermore, mobile device application use and the popularity of individual applications changes over time, so VGI sources may not consistently reflect on-site activities [38,67]. Given the increasing hurdles raised by VGI companies to access these data and the temporary nature of use trends in mobile device application activity [42], practitioners should consider the relative return on investment of acquiring many different VGI sources [68]. To address some challenges, such as disease monitoring or hazard mapping, managers have developed dedicated VGI platforms to address a specific question and minimize data processing time [69,70]. While such platforms could aid in lake visitation monitoring, this approach is dependent on recruiting and maintaining active VGI platform users. Location data derived from passive background location sharing on cellular devices rather than active posts to specific VGI sources may address some of the biases of traditional VGI data, although these

data also have known biases [52,71], and can be prohibitively expensive to acquire [72]. Future research could explicitly calculate the cost (i.e., necessary personnel and monetary investment) of collecting different VGI sources relative to empirical data collection methods. In light of the uncertainty surrounding long-term research access to some platforms (e.g., Twitter) and VGI's ability to simply enhance but not replace on-site estimates [38,73], collection of on-site visitation and survey data will likely remain critical for recreational site management [33].

Broadly, the importance of amenities in driving lake visitation suggests that if managers and policymakers seek to enhance lake access and use then they should invest in improvement of and additional lakeside facilities. Notably, the allocation of urban waters and parks is spatially inequitable [71], and this inequity is further compounded by limited water access and amenities at some lakes. In western Washington, for example, lakes in wealthier suburban neighborhoods typically enjoy superior amenities and access compared to lakes in urban regions of lower socioeconomic status. Equitable enhancement of lake access will be best supported through investing in the built lakeside environment, but this should not be prioritized at the expense of retaining some natural lakeside habitats, as lakes are also valued as settings to connect with nature [74]. Practitioners face tradeoffs between enhancing built infrastructure that supports some forms of recreation while potentially diminishing nature-based recreation at the same time [75]. Plans to increase access to or enhance visitor infrastructure at blue spaces should acknowledge that different cultural values and worldviews may lead to differing preferences and seek ways to address existing inequities in access to different types of lake environments [76].

## Conclusion

VGI reflect relative differences in visitation between lakes and can be used to help estimate visitation at sites lacking detailed on-site visitor count data. Widely used VGI sources such as Twitter and Gaia GPS appear to be moderate predictors of lake visitation in western Washington, where VGI tailored toward niche activities such as eBird and iNaturalist provide additional, albeit minor, contributions in a visitation model. While this may be viewed as a limited return on investment for adding additional VGI datasets to visitation models, we caution researchers to consider the inherent biases of different mobile device applications and how each VGI source may reflect only portions of the suite of CES that are provided by a location. Simply put, diverse VGI sources are likely to characterize the diversity of reasons motivating people to interact with nature. Ultimately, our analysis reinforces the need for quality empirical data which data from mobile devices can complement. VGI cannot fully substitute for on-site data but can enhance visitation models informed by both VGI and on-site data to guide lake and visitor management.

## Author contributions

**Conceptualization:** Rachel M. Fricke, Spencer A. Wood, Julian D. Olden.

**Data curation:** Rachel M. Fricke, Spencer A. Wood.

**Formal analysis:** Rachel M. Fricke, Spencer A. Wood.

**Funding acquisition:** Rachel M. Fricke, Julian D. Olden.

**Methodology:** Rachel M. Fricke, Spencer A. Wood.

**Project administration:** Julian D. Olden.

**Resources:** Spencer A. Wood, Julian D. Olden.

**Supervision:** Julian D. Olden.

**Visualization:** Rachel M. Fricke.

**Writing – original draft:** Rachel M. Fricke.

**Writing – review & editing:** Spencer A. Wood, Julian D. Olden.

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
