## [Decision Letter · Decision Letter 0]

6 Oct 2025

Dear Dr. Fricke,

Thank you for submitting your manuscript to PLOS ONE. After careful consideration, we feel that it has merit but does not fully meet PLOS ONE’s publication criteria as it currently stands. Therefore, we invite you to submit a revised version of the manuscript that addresses the points raised during the review process.

We look forward to receiving your revised manuscript.

Kind regards,

Kealeboga Kaizer Moreri, PhD

Academic Editor

PLOS ONE

Journal Requirements:

2. In your Methods section, please include additional information about your dataset and ensure that you have included a statement specifying whether the collection and analysis method complied with the terms and conditions for the source of the data.

3. Please note that PLOS One has specific guidelines on code sharing for submissions in which author-generated code underpins the findings in the manuscript. In these cases, we expect all author-generated code to be made available without restrictions upon publication of the work. Please review our guidelines at https://journals.plos.org/plosone/s/materials-and-software-sharing#loc-sharing-code and ensure that your code is shared in a way that follows best practice and facilitates reproducibility and reuse.

“This work was supported by a U.S. Geological Survey Northwest Climate Adaptation Science Center award G17AC00218 to Rachel Fricke, the Future Rivers program at the University of Washington as part of a NSF National Research Traineeship award (DGE 1922004), and the University of Washington eScience Institute.”

5. We note that you have indicated that there are restrictions to data sharing for this study. PLOS only allows data to be available upon request if there are legal or ethical restrictions on sharing data publicly. For more information on unacceptable data access restrictions, please see http://journals.plos.org/plosone/s/data-availability#loc-unacceptable-data-access-restrictions.

6. We note that Figures 1 & 2 in your submission contain [map/satellite] images which may be copyrighted. All PLOS content is published under the Creative Commons Attribution License (CC BY 4.0), which means that the manuscript, images, and Supporting Information files will be freely available online, and any third party is permitted to access, download, copy, distribute, and use these materials in any way, even commercially, with proper attribution. For these reasons, we cannot publish previously copyrighted maps or satellite images created using proprietary data, such as Google software (Google Maps, Street View, and Earth). For more information, see our copyright guidelines: http://journals.plos.org/plosone/s/licenses-and-copyright.

a. You may seek permission from the original copyright holder of Figures 1 & 2 to publish the content specifically under the CC BY 4.0 license.

Reviewers' comments:

Reviewer's Responses to Questions

**Comments to the Author**

1. Is the manuscript technically sound, and do the data support the conclusions?

Reviewer #1: Partly

Reviewer #2: Yes

2. Has the statistical analysis been performed appropriately and rigorously?

Reviewer #1: Yes

Reviewer #2: Yes

3. Have the authors made all data underlying the findings in their manuscript fully available?

Reviewer #1: No

Reviewer #2: Yes

4. Is the manuscript presented in an intelligible fashion and written in standard English?

Reviewer #1: Yes

Reviewer #2: Yes

Reviewer #1: I have read the revised version of the manuscript as well as the authors’ responses to the comments. In many cases, I am convinced by the revisions; however, I still have some additional comments that I believe could further improve the work.

I found myself a bit confused about the model and the modeling process. Was the main aim only to explore the relationship between social media–recorded visits and actual on-site visitation? If so, what is the source of the real visitation data (on-site visitation), and is it publicly accessible? I also did not clearly understand what is meant by the “combined R².” The reported value of 0.85 is mentioned in the text but does not appear in Table 1.

It also seems that the relationship between visitation and its influencing factors has been modeled, with results suggesting that “built lakeside infrastructure” is the most important factor. However, the explanations regarding the modeling approach, data preparation, regression coefficients, significance tests, and similar details are either too brief or missing altogether.

Given that in some applications—such as wildlife disease monitoring or risks like building fires—dedicated VGI systems have been designed, and these can provide many advantages, I suggest the authors discuss in the Discussion section whether such a capacity might also exist in the future for monitoring lake visitation. For example, in a dedicated VGI platform, data are collected specifically to answer a targeted question, which reduces the need for extensive data preprocessing. Moreover, mechanisms for quality control (e.g., expert review or credibility scoring for active contributors) can be embedded into the system.

Some relevant examples of dedicated VGI platforms are as follows:

• Di Lorenzo, A., Zenobio, V., Cioci, D., Dall’Acqua, F., Tora, S., Iannetti, S., ... & Di Sabatino, D. (2023). A web-based geographic information system monitoring wildlife diseases in Abruzzo and Molise regions, Southern Italy. BMC Veterinary Research, 19(1), 183.

• Vahidnia, M. H., Hosseinali, F., & Shafiei, M. (2020). Crowdsource mapping of target buildings in hazard: The utilization of smartphone technologies and geographic services. Applied Geomatics, 12(1), 3–14.

Reviewer #2: Interesting subject, but not a new one, as there are several approaches about using VGI to study visitation patterns. The answers provided to reviewer #1 are correct and adequate to the questions raised.

My only recommendation is to update the references used, as the most recent date is from 2023.

**Do you want your identity to be public for this peer review?** For information about this choice, including consent withdrawal, please see our Privacy Policy

Reviewer #1: No

Reviewer #2: No

---

## [Author Response · Author response to Decision Letter 1]

23 Dec 2025

Journal Requirements:

We have edited file names and document formatting to meet these requirements.

2. In your Methods section, please include additional information about your dataset and ensure that you have included a statement specifying whether the collection and analysis method complied with the terms and conditions for the source of the data.

We have added the following statement to our Methods and are happy to provide additional information about the dataset with specific guidance: “All VGI data collection and analysis methods complied with the terms and conditions for each respective data source.” (l. 208-209)

3. Please note that PLOS One has specific guidelines on code sharing for submissions in which author-generated code underpins the findings in the manuscript. In these cases, we expect all author-generated code to be made available without restrictions upon publication of the work. Please review our guidelines at https://journals.plos.org/plosone/s/materials-and-software-sharing#loc-sharing-code and ensure that your code is shared in a way that follows best practice and facilitates reproducibility and reuse.

We have made our code and the data we are permitted to share available on Zenodo (link: 10.5281/zenodo.17654557).

“This work was supported by a U.S. Geological Survey Northwest Climate Adaptation Science Center award G17AC00218 to Rachel Fricke, the Future Rivers program at the University of Washington as part of a NSF National Research Traineeship award (DGE 1922004), and the University of Washington eScience Institute.”

We have provided an amended Funding Statement in the cover letter.

5. We note that you have indicated that there are restrictions to data sharing for this study. PLOS only allows data to be available upon request if there are legal or ethical restrictions on sharing data publicly. For more information on unacceptable data access restrictions, please see http://journals.plos.org/plosone/s/data-availability#loc-unacceptable-data-access-restrictions.

We have edited our data availability statement and provide a revised version below indicating that all shareable datasets are now available on Zenodo:

“All shareable data underlying the results presented in this study are available on Zenodo [10.5281/zenodo.17654557]. VGI data cannot be shared publicly because the data are owned by third parties and our terms of use do not allow us to share the data. VGI data related to this study, including volunteered geographic information records from Flickr, Twitter (now X), Gaia GPS, eBird, and iNaturalist, are available upon download or request from the following webpages and application programming interfaces (APIs): Flickr [https://www.flickr.com/services/developer/api/], eBird [https://science.ebird.org/en/use-ebird-data/download-ebird-data-products], iNaturalist [doi: 10.15468/ab3s5x], and Twitter (X) [https://developer.x.com/en/docs/x-api]. Please note that the free Academic API the authors used to acquire Twitter (X) data is now deprecated under new company policies. Researchers interested in accessing Gaia GPS data must submit a formal data access request directly to the company outlining their research goals and ensuring compliance with relevant privacy regulations [https://www.gaiagps.com/].”

6. We note that Figures 1 & 2 in your submission contain [map/satellite] images which may be copyrighted. All PLOS content is published under the Creative Commons Attribution License (CC BY 4.0), which means that the manuscript, images, and Supporting Information files will be freely available online, and any third party is permitted to access, download, copy, distribute, and use these materials in any way, even commercially, with proper attribution. For these reasons, we cannot publish previously copyrighted maps or satellite images created using proprietary data, such as Google software (Google Maps, Street View, and Earth). For more information, see our copyright guidelines: http://journals.plos.org/plosone/s/licenses-and-copyright.

a. You may seek permission from the original copyright holder of Figures 1 & 2 to publish the content specifically under the CC BY 4.0 license.

We have edited Figures 1 and 2 to use freely available US Census Bureau basemaps instead, given GADM’s limitations on dissemination for commercial use.

We have also updated the figure captions:

“Fig 1. Locations of 50 lakes in Western Washington, United States. The basemap is freely available from the US Census Bureau.” (l. 157-158)

“Fig 2. Map of cumulative VGI user-days at 50 lakes in Western Washington for 2015-2019 on eBird (dark blue), iNaturalist (teal), Flickr (yellow), Twitter (orange), and Gaia (red). Point size corresponds to the number of user-days and point locations are jittered to ease interpretation. The basemap is freely available from the US Census Bureau.” (l. 328-331)

We have reviewed our reference list and formatting and attest that it is complete and correct. We do not cite any retracted articles.

Reviewer #1:

I have read the revised version of the manuscript as well as the authors’ responses to the comments. In many cases, I am convinced by the revisions; however, I still have some additional comments that I believe could further improve the work.

We thank the reviewer for their affirmation of our efforts to address previous reviewers’ comments and their suggestions to further strengthen our manuscript. We have endeavored to address all of their comments through our revisions described below.

I found myself a bit confused about the model and the modeling process. Was the main aim only to explore the relationship between social media–recorded visits and actual on-site visitation? If so, what is the source of the real visitation data (on-site visitation), and is it publicly accessible? I also did not clearly understand what is meant by the “combined R².” The reported value of 0.85 is mentioned in the text but does not appear in Table 1.

Yes, the main aim of our manuscript was to examine the relationship between visitation estimates derived from different combinations of VGI data sources and on-site visitation. On-site visitation data comes from instantaneous counts collected by county volunteers, as described in the “On-site Lake Visitation” subsection of the Methods. The data is available in our public Zenodo repository [10.5281/zenodo.17654557]. “Combined R²” refers to the proportion of variance in on-site visitation that is explained by the all VGI sources model (e.g.,in-sample testing), whereas the R² values in Table 1 are from out-of-sample testing of candidate models (as described in the table caption). We have further clarified this by editing the abstract text, where in-sample was not previously specified: “All VGI sources were included in the top-performing visitation models, suggesting they provide significant and unique contributions to estimates of overall lake use (combined R2 = 0.85, in-sample testing)” (l. 35-37).

It also seems that the relationship between visitation and its influencing factors has been modeled, with results suggesting that “built lakeside infrastructure” is the most important factor. However, the explanations regarding the modeling approach, data preparation, regression coefficients, significance tests, and similar details are either too brief or missing altogether.

We apologize for not offering a sufficient description of our methods in the original submission. We have rectified this omission by editing the “Revealed Preference Model” subsection of the Methods to further clarify these details: “We modeled visitation estimates as a function of lake amenities, water quality, and shoreline tree cover to assess the associations between measures of lake attractiveness and degree of human use. Visitation estimates came from the previously described visitation model which predicted visitation from all VGI sources at all 50 lakes for the five years of our study, thus also including years during which we lacked on-site data at certain lakes. Correlated lake attributes related to the presence of amenities (parks, bathrooms, shelters, public docks, playgrounds, and swimming beaches) were aggregated into a single lakeside amenity variable based on each of these predictors having a variance inflation factor (VIF) greater than 2.5. These six significantly correlated general amenities were combined into a lake amenity score metric with a range from one to six, while boat ramps were included as a separate presence/absence variable to reflect recreation specifically for boating and fishing. Ultimately, our linear mixed-effects model included the combined built lakeside infrastructure variable, boat ramps, water quality, shoreline tree cover, and a random lake effect as predictor variables, with visitation estimates as the response variable. All statistical analyses were completed using the lme4 package in R version 4.3.1 [47].” (l. 283-301)

Given that in some applications—such as wildlife disease monitoring or risks like building fires—dedicated VGI systems have been designed, and these can provide many advantages, I suggest the authors discuss in the Discussion section whether such a capacity might also exist in the future for monitoring lake visitation. For example, in a dedicated VGI platform, data are collected specifically to answer a targeted question, which reduces the need for extensive data preprocessing. Moreover, mechanisms for quality control (e.g., expert review or credibility scoring for active contributors) can be embedded into the system.

Some relevant examples of dedicated VGI platforms are as follows:

• Di Lorenzo, A., Zenobio, V., Cioci, D., Dall’Acqua, F., Tora, S., Iannetti, S., ... & Di Sabatino, D. (2023). A web-based geographic information system monitoring wildlife diseases in Abruzzo and Molise regions, Southern Italy. BMC Veterinary Research, 19(1), 183.

• Vahidnia, M. H., Hosseinali, F., & Shafiei, M. (2020). Crowdsource mapping of target buildings in hazard: The utilization of smartphone technologies and geographic services. Applied Geomatics, 12(1), 3–14.

This is an excellent suggestion. We have made the following additions to our Discussion to address this comment:

“To address some challenges, such as disease monitoring or hazard mapping, managers have developed dedicated VGI platforms to address a specific question and minimize data processing time [69,70]. While such platforms could aid in lake visitation monitoring, this approach is dependent on recruiting and maintaining active VGI platform users. Location data derived from passive background location sharing on cellular devices rather than active posts to specific VGI sources may address some of the biases of traditional VGI data, thou

---

## [Editor Report · Decision Letter 1]

13 Jan 2026

Multiple sources of volunteered geographic information strengthen holistic estimates of lake visitation

PONE-D-25-41499R1

Dear Dr. Fricke,

We’re pleased to inform you that your manuscript has been judged scientifically suitable for publication and will be formally accepted for publication once it meets all outstanding technical requirements.

Kind regards,

Kealeboga Kaizer Moreri, PhD

Academic Editor

PLOS One
---

## [Editor Report · Acceptance letter]

PONE-D-25-41499R1

PLOS One

Dear Dr. Fricke,

I'm pleased to inform you that your manuscript has been deemed suitable for publication in PLOS One. Congratulations! Your manuscript is now being handed over to our production team.

Kind regards,

on behalf of

Dr. Kealeboga Kaizer Moreri

Academic Editor

PLOS One